# Counterfactual Supervision-Based Information Bottleneck for Out-of-Distribution Generalization

**DOI:** 10.3390/e25020193

**Published:** 2023-01-18

**Authors:** Bin Deng, Kui Jia

**Affiliations:** School of Electronic and Information Engineering, South China University of Technology, Guangzhou 510641, China

**Keywords:** out-of-distribution generalization, information bottleneck, causal learning

## Abstract

Learning invariant (causal) features for out-of-distribution (OOD) generalization have attracted extensive attention recently, and among the proposals, invariant risk minimization (IRM) is a notable solution. In spite of its theoretical promise for linear regression, the challenges of using IRM in linear classification problems remain. By introducing the information bottleneck (IB) principle into the learning of IRM, the IB-IRM approach has demonstrated its power to solve these challenges. In this paper, we further improve IB-IRM from two aspects. First, we show that the key assumption of support overlap of invariant features used in IB-IRM guarantees OOD generalization, and it is still possible to achieve the optimal solution without this assumption. Second, we illustrate two failure modes where IB-IRM (and IRM) could fail in learning the invariant features, and to address such failures, we propose a *Counterfactual Supervision-based Information Bottleneck (CSIB)* learning algorithm that recovers the invariant features. By requiring counterfactual inference, CSIB works even when accessing data from a single environment. Empirical experiments on several datasets verify our theoretical results.

## 1. Introduction

Modern machine learning models are prone to catastrophic performance loss during deployment when the test distribution is different from the training distribution. This phenomenon has been repeatedly witnessed and intentionally exposed in many examples [1,2,3,4,5]. Among the explanations, shortcut learning [6] is considered as a main factor causing this phenomenon. A good example is the classification of images of cows and camels—a trained convolutional network tends to recognize cows or camels by learning spurious features from image backgrounds (e.g., green pastures for cows and deserts for camels), rather than learning the causal shape features of the animals [7]; decisions based on the spurious features would make the learned models fail when cows or camels appear in unusual or different environments. Machine learning models are expected to have the capability of out-of-distribution (OOD) generalization and avoid shortcut learning.

To achieve OOD generalization, recent theories [8,9,10,11,12] are motivated by causality literature [13,14] and resort to extraction of the invariant, causal features and establishing the relevant conditions under which machine learning models have guaranteed generalization. Among these works, invariant risk minimization (IRM) [8] is a notable learning paradigm that incorporates the invariance principle [15] into practice. In spite of the theoretical promise of IRM, it is only applicable to problems of linear regression. For other problems, such as linear classification, Ahuja et al. [12] first show that for OOD generalization, linear classification is more difficult (see Theorem 1) and propose a new learning method of information bottleneck-based invariant risk minimization (IB-IRM) based on the support overlap assumption (Assumption 7). In this work, we closely investigate the conditions identified in [12] and propose improved results for OOD generalization of linear classification.

Our technical contributions are as follows. In [12], a notion of support overlap of invariant features is assumed in order to make the OOD generalization of linear classification successful. In this work, we first show that this assumption is strong, but it is still possible to achieve such goal without this assumption. Then, we examine whether the IB-IRM proposed in [12] is sufficient to learn invariant features for linear classification and find that IB-IRM (and IRM) could fail in two modes. We then analyze two failure modes of IB-IRM and IRM, in particular when the spurious features in training environments capture sufficient information for the task of interest but have less information than the invariant features. Based on the above analyses, we propose a new method, termed counterfactual supervision-based information bottleneck (CSIB), to address such failures. We prove that without the need of the support overlap assumption, CSIB is theoretically guaranteed for the success of OOD generalization in linear classification. Notably, CSIB works even when accessing data from a single environment. Finally, we design three synthetic datasets and a colored MINST dataset based on our examples; experiments demonstrate the effectiveness of CSIB empirically.

The rest of this article is organized as follows. The learning problem of out-of-distribution (OOD) generalization is formulated in Section 2. In Section 3, we study the learnability of the OOD generalization with different assumptions to the training and test environments. Using these assumptions, two failure modes of previous methods (IRM and IB-IRM) are analysed in Section 4. Based on the above analysis, our method is then proposed in Section 5. The experiments are reported in Section 6. Finally, we discuss the related works in Section 7 and provide some conclusions and limitations of our work in Section 8. All the proofs and details of experiments are given in the Appendix A and Appendix B.

## 2. OOD Generalization: Background and Formulations

### 2.1. Background on Structural Equation Models

Before introducing our formulations of OOD generalization, we provide a detailed background on structural equation models (SEMs) [8,13].

**Definition 1** (Structural Equation Model (SEM)). *A structural equation model (SEM) C:=(S,N) governing the random vector $X=(X_{1},\ldots ,X_{d})$ is a set of structural equations:*
$$S_{i}:X_{i}\leftarrow f_{i}\left(Pa\left(X_{i}\right),N_{i}\right),$$*where $Pa\left(X_{i}\right)\subseteq \left\{X_{1},\ldots ,X_{d}\right\}\setminus \left\{X_{i}\right\}$ are called the parents of $X_{i}$, and $N_{i}$ are independent noise random variables. For every SEM, we yield a directed acyclic graph (DAG) G by adding one vertex for each $X_{i}$ and directed edges from each parent in $Pa(X_{i})$ (the causes) to child $X_{i}$ (the effect).*

**Definition 2** (Intervention). *Consider an SEM C=(S,N). An intervention e on C consists of replacing one or several of its structural equations to obtain an intervened SEM $C^{e}=\left(S^{e},N^{e}\right)$, with structural equations:*
$$S_{i}^{e}:X_{i}^{e}\leftarrow f_{i}^{e}\left(Pa^{e}\left(X_{i}^{e}\right),N_{i}^{e}\right),$$*The variable $X^{e}$ is intervened if $S_{i}\neq S_{i}^{e}$ or $N_{i}\neq N_{i}^{e}$.*

In an SEM C, we can draw samples from the observational distribution P(X) according to the topological ordering of its DAG G. We can also manipulate (intervene) a unique SEM C in different ways, indexed by *e*, to different but related SEMs $C^{e}$, which results in different interventional distributions $P(X^{e})$. Such family of interventions are used to model the environments.

### 2.2. Formulations of OOD Generalization

In this paper, we study the OOD generalization problem by following the linear classification structural equation model below [12].

**Assumption 1** (Linear classification SEM $C_{ood}$).
(1)$$\begin{array}{cc} \; & Y\leftarrow 1\left(w_{inv}^{*}\cdot Z_{inv}\right)⊕N,\;N∼Bernoulli\left(q\right),\;q<\frac{1}{2};\\ \; & X\leftarrow S(Z_{inv},Z_{spu}), \end{array}$$*where $w_{inv}^{*}\in R^{m}$ is the labeling hyperplane, $Z_{inv}\in R^{m}$, $Z_{spu}\in R^{o}$, $X\in R^{d}$, *⊕* is the XOR operator, $S\in R^{d\times (m+o)}$ is invertible (d=m+o), · is the dot product function, and 1(a)=1 if a≥0 otherwise *0*.*

The SEM $C_{ood}$ governs four random variables $\left\{X,Y,Z_{inv},Z_{spu}\right\}$, and its directed acyclic graph (DAG) is illustrated in Figure 1a, where the exogenous noise variable *N* is omitted. Following Definition 2, each intervention *e* generates a new environment *e* with interventional distribution $P(X^{e},Y^{e},Z_{inv}^{e},Z_{spu}^{e})$. We assume only the variables of $X^{e}$ and $Y^{e}$ are observable. In OOD generalization, we are interested in a set of environments $E_{all}$ defined as below.

**Definition 3** ($E_{all}$). *Consider the SEM $C_{ood}$ (Assumption 1) and the learning goal of predicting Y from X. Then, the set of all environments $E_{all}\left(C_{ood}\right)$ indexes all the interventional distributions $P(X^{e},Y^{e})$ obtainable by valid interventions e. An intervention $e\in E_{all}\left(C_{ood}\right)$ is valid as long as (i) the DAG remains acyclic, (ii) $P\left(Y^{e}|Z_{inv}^{e}\right)=P\left(Y|Z_{inv}\right)$, and (iii) $P\left(X^{e}|Z_{inv}^{e},Z_{spu}^{e}\right)=P\left(X|Z_{inv},Z_{spu}\right)$.*

Assumption 1 shows that $Z_{inv}$ is the cause of the response *Y*. We name $Z_{inv}$ the invariant features or causal features because $P\left(Y^{e}|Z_{inv}^{e}\right)=P\left(Y|Z_{inv}\right)$ always holds among all valid interventional SEMs $C_{ood}^{e}$, as defined in Definition 3. The $Z_{spu}$ is called spurious features because $P(Y^{e}|Z_{spu}^{e})$ may vary in different environments of $E_{all}$.

Let $D={\left\{D^{e}\right\}}_{e\in E_{tr}}$ be the training data gathered from a set of training environments $E_{tr}\subset E_{all}$, where $D^{e}={\left\{\left(x_{i}^{e},y_{i}^{e}\right)\right\}}_{i=1}^{n_{e}}$ is the dataset from environment *e* with each instance $\left(x_{i}^{e},y_{i}^{e}\right)$ i.i.d. drawn from $P(X^{e},Y^{e})$. Let $X^{e}\subseteq R^{d}$ and Y⊆{0,1} be the support sets of $X^{e}$ and *Y*, respectively. Given observed data *D*, the goal of OOD generalization is to find a predictor $f:R^{d}\to Y$ such that it can perform well across a set of OOD environments (test environments) $E_{ood}$ of interest, where $E_{ood}\subseteq E_{all}$. Formally, it is expected to minimize
(2)$$\max_{e\in E_{ood}}R^{e}\left(f\right),$$
where $R^{e}\left(f\right):=E_{X^{e},Y^{e}}\left[l\left(f\left(X^{e}\right),Y^{e}\right)\right]$ is the risk under the environment *e* with l(·,·) the 0-1 loss function. Since $E_{ood}$ may be different from $E_{tr}$, this learning problem is called OOD generalization. We assume the predictor f=w∘Φ includes a feature extractor Φ:X→H and a classifier w:H→Y. With a slight abuse of notation, we also let the classifier *w* and feature extractor Φ be parameteried by themselves, respectively, as $w\in R^{c+1}$ and $\Phi \in R^{c\times d}$ with *c* the number of feature dimension.

### 2.3. Background on IRM and IB-IRM

To minimize Equation (Equation 2), two notable solutions of IRM [8] and IB-IRM [12] are listed as follows:(3)$$\mathrm{IRM}:\;\min_{w,\Phi }\frac{1}{|E_{tr}|}\sum_{e\in E_{tr}}R^{e}\left(w\circ \Phi \right),\;s.t.\;w\in \arg\min_{\tilde{w}}R^{e}\left(\tilde{w}\circ \Phi \right),\forall e\in E_{tr},$$
(4)$$\mathrm{IB}-\mathrm{IRM}:\;\min_{w,\Phi }\sum_{e\in E_{tr}}h^{e}\left(\Phi \right),\;s.t.\;\frac{1}{|E_{tr}|}\sum_{e\in E_{tr}}R^{e}\left(w\circ \Phi \right)\leq rth,w\in \arg\min_{\tilde{w}}R^{e}\left(\tilde{w}\circ \Phi \right),\forall e\in E_{tr},$$
where $R^{e}\left(w\circ \Phi \right)=E_{X^{e},Y^{e}}\left[l\left(w\circ \Phi \left(X^{e}\right),Y^{e}\right)\right]$, and $h^{e}\left(\Phi \right)=H\left(\Phi \left(X^{e}\right)\right)$ with *H* the Shannon entropy (or a lower bounded differential entropy), and rth is the threshold on the average risk. If we drop the invariance constraint from IRM and IB-IRM, we obtain standard empirical risk minimization (ERM) and information bottleneck-based empirical risk minimization (IB-ERM), respectively. The use of an entropy constraint in IB-IRM is inspired from the information bottleneck principle [16] where mutual information I(X;Φ(X)) is used for information compression. Since the representation Φ(X) is a deterministic mapping of *X*, we have
(5)I(X;Φ(X))=H(Φ(X))−H(Φ(X)|X)=H(Φ(X)),
thus minimizing the entropy of Φ(X) is equivalent to minimizing the mutual information I(X;Φ(X)). In brief, the optimization goal of IB-IRM is to select the one that has the least entropy among all highly predictive invariant predictors.

## 3. OOD Generalization: Assumptions and Learnability

To study the learnability of OOD generalization, we make following definition.

**Definition 4.** 
*Given $E_{tr}\subset E_{all}$ and $E_{ood}\subseteq E_{all}$. We say an algorithm succeeds to solve OOD generalization with respect to ($E_{tr},E_{ood}$) if the predictor $f^{*}\in F$ returned by this algorithm satisfies the following equation:*

(6)
$$\max_{e\in E_{ood}}R^{e}\left(f^{*}\right)=\min_{f\in F}\max_{e\in E_{ood}}R^{e}\left(f\right),$$

*where F is the learning hypothesis (a function set including all possible linear classifier). Otherwise we say it fails to solve OOD generalization.*


So far, we have omitted how different environments of $E_{tr}$ and $E_{ood}$ exactly are to enable OOD generalization. Different assumptions about $E_{tr}$ and $E_{ood}$ make the OOD generalization problem different.

### 3.1. Assumptions about the Training Environments $E_{tr}$

Define the support set of the invariant (*resp.*, spurious) features $Z_{inv}^{e}$ (*resp.*, $Z_{spu}^{e}$) in environment *e* as $Z_{inv}^{e}$ (*resp.*, $Z_{spu}^{e}$). In general, we make following assumptions to the invariant features $Z_{inv}^{e}$ in the training environments $E_{tr}$.

**Assumption 2** (Bounded invariant features). *$\cup_{e\in E_{tr}}Z_{inv}^{e}$ is a bounded set. (A set Z is bounded if ∃M<∞ such that ∀z∈Z,∥z∥≤M).*

**Assumption 3** (Strictly separable invariant features). *$\forall z\in \cup_{e\in E_{tr}}Z_{inv}^{e},w_{inv}^{*}\cdot z\neq 0.$*

The difficulties of OOD generalization are due to the spurious correlations between $Z_{inv}$ and $Z_{spu}$ in the training environments $E_{tr}$. In this paper, we consider three modes induced by different correlations between $Z_{inv}$ and $Z_{spu}$ as shown below.

**Assumption 4** (Spurious correlation 1). *Assume each $e\in E_{tr}$,*
(7)$$\begin{array}{cc} \; & Z_{spu}^{e}\leftarrow AZ_{inv}^{e}+W^{e}; \end{array}$$*where $A\in R^{o\times m}$, and $W^{e}\in R^{o}$ is a continuous (or discrete with each component supported on at least two distinct values), bounded, and zero mean noise variable.*

**Assumption 5** (Spurious correlation 2). *Assume each $e\in E_{tr}$,*
(8)$$\begin{array}{cc} \; & Z_{inv}^{e}\leftarrow AZ_{spu}^{e}+W^{e}; \end{array}$$*where $A\in R^{m\times o}$, and $W^{e}\in R^{m}$ is a continuous (or discrete with each component supported on at least two distinct values), bounded, and zero mean noise variable.*

**Assumption 6** (Spurious correlation 3). *Assume each $e\in E_{tr}$,*
(9)$$\begin{array}{cc} \; & Z_{spu}^{e}\leftarrow W_{1}^{e}Y^{e}+W_{0}^{e}\left(1-Y^{e}\right); \end{array}$$*where $W_{0}^{e}\in R^{o}$ and $W_{1}^{e}\in R^{o}$ are independent noise variables.*

For each $e\in E_{tr}$, the DAGs of its corresponding interventional SEMs $C_{ood}^{e}$ with respect to Assumptions 4–6 are illustrated in Figure 1b–d, respectively. It is worth noting that although the DAGs are identical across all training environments in each mode of Assumptions 4–6, the interventional SEMs $C_{ood}^{e}$ among different training environments are different due to the interventions on the exogenous noise variables.

### 3.2. Assumptions about the OOD Environments $E_{ood}$

**Theorem 1** (Impossibility of guaranteed OOD generalization for linear classification [12]). *Suppose $E_{ood}=E_{all}$. If for all the training environments $E_{tr}$, the latent invariant features are bounded and strictly separable, i.e., Assumptions 2 and 3 hold, then every deterministic algorithm fails to solve the OOD generalization.*

The above theorem shows that it is impossible to solve OOD generalization if $E_{ood}=E_{all}$. To make it learnable, Ahuja et al. [12] propose the support overlap assumption (Assumption 7) to the invariant features.

**Assumption 7** (Invariant feature support overlap). *$\forall e\in E_{ood},Z_{inv}^{e}\subseteq \cup_{e^{\prime }\in E_{tr}}Z_{inv}^{e^{\prime }}$.*

However, Assumption 7 is strong, and we would show that it is still possible to solve OOD generalization without this assumption. For better illustration, consider an OOD generalization task from $P(X^{e_{1}},Y^{e_{2}})$ to $P(X^{e_{2}},Y^{e_{2}})$ with $E_{tr}=\left\{e_{1}\right\}$ and $E_{ood}=\left\{e_{2}\right\}$, and the support sets of the corresponding invariant features $Z_{inv}^{e_{1}}$ and $Z_{inv}^{e_{2}}$ are intuitively illustrated in Figure 2c (assume $dim(Z_{inv})=2$ in this example). From Figure 2c, it is clear that although the support sets of invariant features between the two environments are different, it is still possible to solve OOD generalization if the learned feature extractor Φ only captures the invariant features, e.g., $\Phi \left(X\right)=Z_{inv}$.

To make Assumption 7 weaker, we propose the following assumption.

**Assumption 8.** 
*Let $P\left(Z_{inv}^{tr},Y^{tr}\right)=\frac{1}{|E_{tr}|}\sum_{e\in E_{tr}}P\left(Z_{inv}^{e},Y^{e}\right)$ be the mixture distribution of invariant features in the training environments. Denote A be a hypothesis set including all linear classifiers mapping from $R^{m}$ to Y. $\forall e\in E_{ood}$, assume $F_{l}\left(P\left(Z_{inv}^{tr},Y^{tr}\right)\right)\subseteq F_{l}\left(P\left(Z_{inv}^{e},Y^{e}\right)\right)$, where l is the 0-1 loss function and $F_{l}\left(P\left(Z,Y\right)\right)=\arg\min_{f\in A}E_{Z,Y}\left[l\left(f\left(Z\right),Y\right)\right]$.*


Clearly, under the assumption of separable invariant features (Assumption 3), for any $e\in E_{ood}$, Assumption 7 holds ⇒$Z_{inv}^{e}\subseteq Z_{inv}^{tr}$⇒$F_{l}\left(P\left(Z_{inv}^{tr},Y^{tr}\right)\right)\subseteq F_{l}\left(P\left(Z_{inv}^{e},Y^{e}\right)\right)$⇒. Assumption 8 holds, but not vice versa. Therefore, Assumption 8 is weaker than Assumption 7. We show that Assumption 8 could be substituted for Assumption 7 for the success of OOD generalization in our proposed method in Section 5.

## 4. Failures of IRM and IB-IRM

Under Spurious Correlation 1 (Assumption 4), the IB-IRM algorithm has been shown to enable OOD generalization, while IRM fails [12]. In this section, we would show that both IRM and IB-IRM could fail under Spurious Correlations 2 and 3 (Assumptions 5 and 6).

### 4.1. Failure under Spurious Correlation 2

**Example 1** (Counter-Example 1). *Under Assumption 5, let $Z_{inv}^{e}\leftarrow Z_{spu}^{e}+W^{e}$ with $dim(Z_{inv}^{e})$$=dim\left(Z_{spu}^{e}\right)=dim\left(W^{e}\right)=1$ and $w_{inv}^{*}=1$ be the generated classifier in Assumption 1. We assume two training environments and a OOD environment as:*
$$\begin{array}{c} E_{tr}=\left\{e_{1},e_{2}\right\};\;E_{ood}=\left\{e_{3}\right\};\\ e_{1}:P\left(Z_{spu}^{e_{1}}=-2\right)=1,P\left(W^{e_{1}}=-1\right)=0.5,P\left(W^{e_{1}}=1\right)=0.5;\\ e_{2}:P\left(Z_{spu}^{e_{2}}=2\right)=1,P\left(W^{e_{2}}=-1\right)=0.5,P\left(W^{e_{2}}=1\right)=0.5;\\ e_{3}:P\left(Z_{spu}^{e_{3}}=1\right)=1,P\left(W^{e_{3}}=-2\right)=0.5,P\left(W^{e_{3}}=2\right)=0.5. \end{array}$$

Figure 2a shows the support points of these features in the training environments. Then, by applying any algorithm to solve the above example with rth=q, we would obtain a predictor of $f^{*}=w^{*}\circ \Phi^{*}$. Consider the prediction made by this model as (we ignore the classifier bias for convenience)
(10)$$f^{*}\left(X^{e}\right)=f^{*}\left(S\left(Z_{inv}^{e},Z_{spu}^{e}\right)\right)=1\left(\Phi_{inv}^{*}Z_{inv}^{e}+\Phi_{spu}^{*}Z_{spu}^{e}\right).$$It is trivial to show that the $f^{*}$ of $\Phi_{inv}^{*}=0$ and $\Phi_{spu}^{*}=1$ is an invariant predictor across training environments with classification error $R^{e_{1}}=R^{e_{2}}=q$, and it achieves the least entropy of $h^{e}\left(\Phi^{*}\right)=0$ for each training environment *e*. Therefore, it is a solution of IB-IRM and IRM. However, the predictor of $f^{*}$ relies on spurious features and has the test error $R^{e_{3}}=0.5$; thus, it fails to solve the OOD generalization.

### 4.2. Failure under Spurious Correlation 3

**Example 2** (Counter-Example 2). *Under Assumption 6, let $Z_{spu}^{e}\leftarrow W_{1}^{e}Y^{e}+W_{0}^{e}\left(1-Y^{e}\right)$ with $dim\left(Z_{inv}\right)=dim\left(Z_{spu}\right)=dim\left(W_{0}^{e}\right)=dim\left(W_{1}^{e}\right)=1$, $Z_{inv}^{e}$ be a discrete variable supported uniformly on six points {−4,−3,−2,2,3,4} among all environments, and $w_{inv}^{*}=1$ be the generated classifier in Assumption 1. We assume two training environments and a OOD environment as:*
$$\begin{array}{c} E_{tr}=\left\{e_{1},e_{2}\right\};\;E_{ood}=\left\{e_{3}\right\}\\ e_{1}:P\left(W_{0}^{e_{1}}=-1\right)=1,P\left(W_{1}^{e_{1}}=1\right)=1;\\ e_{2}:P\left(W_{0}^{e_{2}}=-0.5\right)=1,P\left(W_{1}^{e_{2}}=0.5\right)=1;\\ e_{3}:P\left(W_{0}^{e_{3}}=1\right)=1,P\left(W_{1}^{e_{3}}=-1\right)=1; \end{array}$$

Figure 2b shows the support points of these features in the training environments. Then, by applying any algorithm to solve the above example with rth=q, we would obtain a predictor of $f^{*}=w^{*}\circ \Phi^{*}$. Consider the prediction made by this model as (we ignore the classifier bias for convenience):(11)$$f^{*}\left(X^{e}\right)=f^{*}\left(S\left(Z_{inv}^{e},Z_{spu}^{e}\right)\right)=1\left(\Phi_{inv}^{*}Z_{inv}^{e}+\Phi_{spu}^{*}Z_{spu}^{e}\right).$$It is trivial to show that the $f^{*}$ of $\Phi_{inv}^{*}=0$ and $\Phi_{spu}^{*}=1$ is an invariant predictor across training environments with classification error $R^{e_{1}}=R^{e_{2}}=0$, and it achieves the least entropy of $h^{e}\left(\Phi^{*}\right)=1$ among all highly predictive predictors for each training environment *e*. and Therefore, it is a solution of IB-IRM and IRM. However, the predictor of $f^{*}$ relies on spurious features and has the test error $R^{e_{3}}=1$; thus, it fails to solve the OOD generalization.

### 4.3. Understanding the Failures

From the illustrations of the above simple examples, we can conclude that the failure of the invariance constraint for removing the spurious features is because the spurious features among all training environments are strictly linearly separable by their corresponding labels. This would make the predictor rely only on spurious features to achieve minimum training error and also be the invariant predictor across training environments. Since the label set is finite (with only two values in binary classification) in classification problems, such a phenomenon may exist. We state such failure mode formally as below.

**Theorem 2.** 
*Given any $E_{tr}\subset E_{all}$ and $E_{ood}\subseteq E_{all}$ satisfying Assumptions 2, 3, and 7, if two sets $\cup_{e\in E_{tr}}Z_{spu}^{e}\left(Y^{e}=1\right)$ and $\cup_{e\in E_{tr}}Z_{spu}^{e}\left(Y^{e}=0\right)$ are linearly separable and $H\left(Z_{inv}^{e}\right)>H\left(Z_{spu}^{e}\right)$ on each training environment e, then IB-IRM (and IRM, ERM, or IB-ERM) with any rth∈R fails to solve the OOD generalization.*


The understanding of Theorem 2 is intuitive since when the spurious features in the training environments with respect to different labels are linearly separable, there is no algorithm that can distinguish spurious features from invariant features. Although the assumption of linear separation of the spurious features seems strong for this failure, it is easy to hold in high-dimensional space when $dim(Z_{spu})$ is large (common cases in practice such as image data). We show one case in Section A.3 that if the number of environments is $|E_{tr}|<dim\left(Z_{spu}\right)/2$ under Assumption 6, the spurious features in the training environments are probably separable by their labels. This is because in *o*-dimensional space there is a high probability that *o* randomly drawn distinct points are linearly separable for any two subsets.

## 5. Counterfactual Supervision-Based Information Bottleneck

In the above analyses, we have shown two failure modes of IB-IRM and IRM for OOD generalization in the linear classification problem. The key reason for the failure is due to the learned features Φ(X) that rely on spurious features. To prevent such failure, we present the counterfactual supervision-based information bottleneck (CSIB) learning algorithm for removing the spurious features progressively.

In general, the IB-ERM method is applied to extract features from the beginning of each iteration:(12)$$\;\min_{w,\Phi }\sum_{e\in E_{tr}}h^{e}\left(\Phi \right)\;s.t.\;\frac{1}{|E_{tr}|}\sum_{e\in E_{tr}}R^{e}\left(w\circ \Phi \right)\leq rth$$Due to the information bottleneck, only a part of the information of the input *X* are exploited in Φ(X). If the information of spurious features $Z_{spu}$ exists in the learned features Φ(X), the idea of CSIB is to drop such information and meanwhile maintain the causal information (represented by invariant features $Z_{inv}$) as well. However, achieving such a goal faces two challenges: (1) how to determine whether Φ(X) contains spurious information of $Z_{spu}$? and (2) how to remove the information of $Z_{spu}$?

Fortunately, due to the orthogonality in the linear space, it is possible to disentangle the features that are exploited by Φ(X) (denoted as $X_{1}$) and the features that are not exploited by Φ(X) (denoted as $X_{2}$) via Singular Value Decomposition (SVD). Based on that, we could construct an SEM $C_{new}$ governing three variables of $X_{1}$, $X_{2}$, and *X*. Therefore, by conducting counterfactual interventions on $X_{1}$ and $X_{2}$ in $C_{new}$, we could solve the first challenge by requiring a single supervision on the counterfactual examples $X^{\prime }$. For example, if we intervene on $X_{1}$ and find that the causal information remains in the resulting $X^{\prime }$, then the extracted features Φ(X) are definitely the spurious features. To address the second challenge, we replace the input by $X_{2}$ by filtering out the information of $X_{1}$ and conduct the same learning procedure from the beginning.

The learning algorithm of CSIB is illustrated in Algorithm 1, and Figure 3 shows the framework of CSIB. We show in Theorem 3 that CSIB is theoretically guaranteed to succeed to solve OOD generalization.
**Algorithm 1** Counterfactual Supervision-based Information Bottleneck (CSIB)**Input:**$P(X^{e},Y^{e})$, $e\in E_{tr}$, rth>0, $c\geq dim(Z_{inv})$, M≫0, and (x,y) is an example randomly drawn from $P(X^{e},Y^{e})$.**Output:** classifier $w\in R^{c+1}$, feature extractor $\Phi =R^{c\times d}$.**Begin:**  1:Lv←[]; Lr←[]; $\Phi^{\prime }\leftarrow I^{d\times d}$  2:$d^{\prime }\leftarrow dim\left(X^{e}\right)$  3:Apply IB-ERM method (Equation (Equation 12)) to $P(X^{e},Y^{e})$ and obtain $w^{*}\in R^{c+1}$ and $\Phi^{*}\in R^{c\times d^{\prime }}$  4:Apply SVD to $\Phi^{*}$ as $\Phi^{*}=U\Lambda V^{T}=\left[U_{1},U_{2}\right]\left[\Lambda_{1},0;0,0\right]\left[V_{1}^{T};V_{2}^{T}\right]$  5:$r\leftarrow rank(\Phi^{*})$  6:$z_{1:r}^{1}\leftarrow \left[-M,\ldots ,-M\right]$; $z_{r+1:d^{\prime }}^{1}\leftarrow V_{2}^{T}\Phi^{\prime }x$  7:$z_{1:r}^{2}\leftarrow \left[M,\ldots ,M\right]$; $z_{r+1:d^{\prime }}^{2}\leftarrow V_{2}^{T}\Phi^{\prime }x$  8:$x^{1}\leftarrow Vz^{1}$; $x^{2}\leftarrow Vz^{2}$  9:**if**Lv is not empty **then**10:    $z_{old}\leftarrow \left[\right]$; i←0; $x^{\prime }\leftarrow x$11:    **while** i<len(Lv) **do**12:        $z\leftarrow Lv\left[i\right]x^{\prime }$13:        $z_{old}$.append(*z*)14:        $x^{\prime }\leftarrow z_{Lr[i]:}$15:        i←i+116:    **end while**17:    i←018:    **while** i<len(Lv) **do**19:        j←len(Lv)−i20:        $z^{1}\leftarrow z_{old}\left[j\right]$; $z^{2}\leftarrow z_{old}\left[j\right]$21:        $z_{Lr[j]:}^{1}\leftarrow x^{1}$; $z_{Lr[j]:}^{2}\leftarrow x^{2}$22:        $x^{1}\leftarrow Lv{\left[j\right]}^{T}z^{1}$; $x^{2}\leftarrow Lv{\left[j\right]}^{T}z^{2}$23:        i←i+124:    **end while**25:**end if**26:**if** label($x^{1}$) = label($x^{2}$) **then**27:    Lr.append(*r*); Lv.append($V^{T}$)28:    $X^{e}\leftarrow V_{2}^{T}X^{e}$; $\Phi^{\prime }\leftarrow V_{2}^{T}\Phi^{\prime }$29:    Go to Step 230:**end if**31:$w\leftarrow w^{*}$; $\Phi \leftarrow \Phi^{*}$**End**

**Theorem 3** (Guarantee of CSIB). *Given any $E_{tr}\subset E_{all}$ and $E_{ood}\subseteq E_{all}$ satisfying Assumptions 2, 3, and 8, then for every spurious correlation of Assumptions 4, 5, and 6 (in this correlation mode, assume the spurious features are linearly separable in the training environments), the CSIB algorithm with rth=q succeeds in solving the OOD generalization.*

**Remark 1.** 
*CSIB succeeds to solve OOD generalization without assuming the support overlap to invariant features and could apply to multiple spurious modes where IB-IRM (as well as ERM, IRM, and IB-ERM) may fail. By introducing counterfactual inference and further supervision (usually conducted by a human) with several steps, CSIB works even when accessing data from a single environment, which is significant especially in the cases where multiple environments’ data are not available.*


## 6. Experiments

### 6.1. Toy Experiments on Synthetic Datasets

We perform experiments on three synthetic datasets from different spurious correlations modes to verify our method—counterfactual, supervision-based, and information bottleneck (CSIB)—and compare them to ERM, IB-ERM, IRM, and IB-IRM. We follow the same protocol for tuning hyperparameters from [8,12,17] and report the classification error for all experiments. In the following, we first briefly describe the designed datasets and then report the main results. More experimental details can be found in the Appendix.

#### 6.1.1. Datasets

**Example 1/1S.** The example is a modified one from the linear unit tests introduced in [17], which generalizes the cow/camel classification task with relevant backgrounds.
$$\begin{array}{cc} \; & \theta_{cow}=1_{m},\;\theta_{camel}=-\theta_{cow},\;\nu_{animal}=10^{-2}\\ \; & \theta_{grass}=1_{o},\;\theta_{sand}=-\theta_{grass},\;\nu_{background}=1. \end{array}$$The dataset $D_{e}$ of each environment $e\in E_{tr}$ is sampled from the following distribution
$$\begin{array}{c} U^{e}∼\mathrm{Categorical}\left(p^{e}s^{e},\left(1-p^{e}\right)s^{e},p^{e}\left(1-s^{e}\right),\left(1-p^{e}\right)\left(1-s^{e}\right)\right),\\ Z_{inv}^{e}∼\left\{\begin{array}{cc} \left(N_{m}\left(0,0.1\right)+\theta_{cow}\right)\nu_{animal} & \;\mathrm{if}\;U^{e}\in \left\{1,2\right\},\\ \left(N_{m}\left(0,0.1\right)+\theta_{camel}\right)\nu_{animal} & \;\mathrm{if}\;U^{e}\in \left\{3,4\right\}, \end{array}\right.\\ Z_{spu}^{e}∼\left\{\begin{array}{cc} \left(N_{o}\left(0,0.1\right)+\theta_{grass}\right)\nu_{background} & \;\mathrm{if}\;U^{e}\in \left\{1,4\right\},\\ \left(N_{o}\left(0,0.1\right)+\theta_{sand}\right)\nu_{background} & \;\mathrm{if}\;U^{e}\in \left\{2,3\right\}, \end{array}\right.\\ Z^{e}\leftarrow \left(Z_{inv}^{e},Z_{spu}^{e}\right),\;X^{e}\leftarrow S\left(Z^{e}\right),\;N∼Bernoulli\left(q\right),\;q<0.5,\;Y^{e}\leftarrow 1\left(1_{m}^{T}Z_{inv}^{e}\right)⊕N \end{array}$$We set $s^{e_{0}}=0.5,s^{e_{1}}=0.7,$ and $s^{e_{2}}=0.3$ for the first three environments, and $s^{e_{j}}∼\mathrm{Uniform}$(0.3,0.7) for j>3. The scrambling matrix *S* is an identical matrix in Example 1 and a random unitary matrix in Example 1S. Here, we set $p^{e}=1$ and q=0 for all environments to make the spurious features and the invariant features both linearly separable to confuse each other. The experiments on different values of *q* and $p^{e}$ are presented in the Appendix, where we have found very interesting observations related to the inductive bias of neural networks.

**Example 2/2S.** This example is extended from Example 1 to show one of the failure modes of IB-IRM (as well as ERM, IRM, and IB-ERM) and how our method can be improved by intervention (counterfactual supervision). Given $w^{e}\in R$, each instance in the environment data $D^{e}$ is sampled by
$$\begin{array}{c} \theta_{spu}=5\cdot 1_{o},\;\theta_{w}=w^{e}\cdot 1_{m},\;\nu_{spu}=10^{-2},\;\nu_{w}=1,\;p,q∼\mathrm{Bernoulli}(0.5),\\ \;Z_{spu}^{e}=N_{o}\left(0,1\right)\nu_{spu}+\left(2p-1\right)\cdot \theta_{spu},\;W^{e}=N_{m}\left(0,1\right)\nu_{w}+\left(2q-1\right)\cdot \theta_{w}\\ Z_{inv}^{e}=AZ_{spu}^{e}+W^{e},\;Z^{e}\leftarrow \left(Z_{inv}^{e},Z_{spu}^{e}\right),\;X^{e}\leftarrow S\left(Z^{e}\right),\;Y^{e}=1\left(1_{m}^{T}Z_{inv}^{e}\right), \end{array}$$
where we set m=o=5, and $A\in R^{m\times o}$ is the identical matrix in our experiments. We set $w^{e_{0}}=3$, $w^{e_{1}}=2$, $w^{e_{2}}=1$, and $w^{e_{j}}=\mathrm{Uniform}\left(0,3\right)$ if j>3 for different training environments. This example shows clearer smaller entropy of spurious features than that of invariant features, which is opposite Example 1/1S.

**Example 3/3S.** This example extends from Example 2 and is similar to the construction of Example 2/2S. Let $w^{e}∼\mathrm{Uniform}\left(0,1\right)$ for different training environments. Each instance in the environments *e* is sampled by
$$\begin{array}{c} \theta_{inv}=\cdot 10\cdot 1_{m},\;\nu_{inv}=10,\;\nu_{spu}=1,\;p,q∼\mathrm{Bernoulli}(0.5),\\ Z_{inv}^{e}=N_{m}\left(0,1\right)\nu_{inv}+\left(2p-1\right)\cdot \theta_{inv},\;Y^{e}=1\left(1_{m}^{T}Z_{inv}^{e}\right),\\ Z_{spu}^{e}=2\left(Y^{e}-1\right)\cdot \nu_{spu}+\left(2q-1\right)\cdot w^{e}\cdot 1_{o},\;Z^{e}\leftarrow \left(Z_{inv}^{e},Z_{spu}^{e}\right),\;X^{e}\leftarrow S\left(Z^{e}\right), \end{array}$$
where we set m=o=5 in our experiments. The spurious features have smaller entropy than the invariant features in this example, which is similar to Example 2/2S, but the invariant features significantly enjoy much larger margin than the spurious features, which is very different from the above two examples. We show a summary of the properties of these three datasets in Table 1 for a general view.

#### 6.1.2. Summary of Results

Table 2 shows the classification errors of different methods when training data comes from single, three, and six environments. We can see that ERM and IRM fail to recognize the invariant features in the experiment of Example 1/1S, where invariant features have smaller margin than spurious features do, while information bottleneck-based methods (IB-ERM, IB-IRM, and CSIB) show improved results due to the smaller entropy of the invariant features. Our method CSIB shows results consistent with IB-IRM in Example 1/1S when invariant features are extracted in the first run, which verifies the effectiveness of using the information bottleneck for OOD generalization. In another dataset of Example 2/2S, where the invariant features have larger entropy than spurious features do, we can see that only CSIB can remove the spurious features compared with the other method, although the information bottleneck-based method IB-ERM would degrade the performance of ERM by focusing more on the spurious features. In the third experiment of Example 3/3S, we can see that although ERM shows not-bad results due to the significantly larger margin of invariant features, our CSIB method still shows improvements by removing more spurious features. Notably, comparing the IB-ERM and IB-IRM when only spurious features are extracted (Example 2/2S, Example 3/3S), our CSIB method could effectively remove them by counterfactual supervision and then refocus on the invariant features. Note that the reason of non-zero average error and the fluctuant results of CSIB in some experiments is that the entropy minimization in the training process is less accurate, where entropy is substituted by variance for the ease of the optimization. Nevertheless, there always exists a case where the entropy is indeed truly minimized, and the error reaches zero (see (min) in the table) in Example 2/2S and Example 3/3S. In summary, CSIB consistently performs better in different spurious correlations modes and is especially more effective than IB-ERM and IB-IRM when the spurious features enjoy much smaller entropy than the invariant features do.

### 6.2. Experiments on Color MNIST Dataset

In this experiment, we set up a binary classification task for digit recognition and identify whether the digit is less than five or more than five. We use real-world dataset, the MNIST database of handwritten digits (http://yann.lecun.com/exdb/mnist/), for the construction. Following our learning setting, we use color information as the spurious features that correlates strongly with the class label. By construction, the label is more strongly correlated with the color than with the digit in the training environments, but this correlation is broken in the test environment. Specifically, the three designed environments (two training environments and one test environment containing 10,000 points each) of the color MNIST are as follows: first, we define a preliminary binary label $\hat{y}$ to the image base on the digit: $\hat{y}=0$ for digits 0–4 and $\hat{y}=1$ for 5–9. Second, we obtain the final label *y* by flipping $\hat{y}$ with probability 0.25. Then, we flip the final labels to obtain the color id, where the flipping probabilities with respect to two training environments and one test environment are 0.2 and 0.1, and 0.9. For better understanding, we randomly draw 20 examples for each label from each environment and visualize them in Figure 4.

The classification results on the color MNIST dataset are shown in Table 3. From the results, we can see that both ERM and IB-ERM methods almost surely use the color features to achieve the task. Although IRM and IB-IRM methods have shown some improvements over ERM, only our method can perform better than a random prediction, which demonstrates the effectiveness of CSIB.

## 7. Related Works

We divide the works related to OOD generalization into two categories: theory and methods, though some of them belong to both.

### 7.1. Theory of OOD Generalization

Based on different definitions to the distributional changes, we review the corresponding theory by the following three categories.

**Based on causality.** Due to the close connection between the distributional changes and the interventions discussed in the theory of causality [13,14], the problem of OOD generalization is usually built in the framework of causal learning. The theory states that a response *Y* is directly caused only by its parents variables $X_{Pa(Y)}$, and all interventions other that those on *Y* do not change the conditional distribution of $P(Y|X_{Pa(Y)})$. Such theory inspires a popular learning principle—the invariance principle—that aims to discover a set of variables such that they remain invariant to the response *Y* in all observed environments [15,19,20]. Invariant risk minimization (IRM) [8] is then proposed to learn a feature extractor Φ in an end-to-end way such that the optimal classifier based on the extracted features Φ(X) remains unchanged in each environment. The theory in [8] shows the guarantee of IRM for OOD generalization under some general assumptions but only focuses on the linear regression tasks. Different from the failure analyses of IRM for the classification tasks in [21,22], where the response Y is the cause of the spurious feature, Ahuja et al. [12] analyse another scenario when the invariant feature is the cause of the spurious feature and show that in this case, linear classification is more difficult than linear regression, where the invariance principle itself is insufficient to ensure the success of OOD generalization. They also claim that the assumption of support overlap of invariant features is necessarily needed. They then propose a learning principle of information bottleneck-based invariant risk minimization (IB-IRM) for linear classification, which shows how to address the failures of IRM by adding information bottleneck [16] into the learning. In this work, we closely investigate the conditions identified in [12] and first show that support overlap of invariant features is not necessarily needed for the success of OOD generalization. We further show several failure cases of IB-IRM and propose improved results for it.

Recently, some works tackle the challenge of OOD generalization in the nonlinear regime [23,24]. Commonly, both of them use variational autoencoder (VAE)-based models [25,26] to identify the latent variables from observations in the first stage. Then, these inferring latent variables are separated into two distinct parts of invariant (causal) and spurious (non-causal) features based on different assumptions. Specifically, Lu et al. [23,27] assume that the latent variables conditioned on some accessible side information such as the environment index or class label follow the exponential family distributions, and Liu et al. [24] directly disentangle the latent variables to two different parts during the inferring stage and assume that the marginal distributions of them are independent of each other. These assumptions, however, are rather strong in general. Nevertheless, these solutions aim to capture the latent variables such that the response given these variables is invariant for different environments, which could still fail because the invariance principle itself is insufficient for OOD generalization in the classification tasks, as shown in [12]. In this work, we focus on the linear classification only and show a new theory of a new method that addresses several OOD generalization failures in the linear settings. Our method could extend to the nonlinear regime by combining with the disentangled representation learning [28] or causal representation learning [29]. Specifically, once the latent representations are well disentangled, i.e., the latent features are represented by a linear transform of the causal features and spurious features, we then could apply our method to filter out the spurious features in the latent space such that only causal features remain.

**Based on robustness.** Different from those based on the causality, where different distributions are generated by intervention on a same SEM and the goal is to discover causal features, the robustness-based methods aim to protect the model against the potential distributional shifts within the uncertainty set, which is usually constrained by f-divergence [30] or Wasserstein distance [31]. This series of works is theoretically addressed by distributionally robust optimization (DRO) under a minimax framework [32,33]. Recently, some works tend to discover the connections between causality and robustness [34]. Although these works show less relevance to us, it is possible that a well-defined measure of distribution divergence could help to effectively extract causal features under the robustness framework. This would be an interesting avenue for future research.

**Others.** Some other works assume that the distributions (domains) are generated from a hyper-distribution and aim to minimize the average risk estimation error bound [35,36,37]. These works are often built based on the generalization theory under the independent and identically distributed (IID) assumption. The authors in [38] do not make any assumption on the distributional changes and only study the learnability of OOD generalization in a general way. All of these theories do not cover the OOD generalization problem under a single training environment or domain.

### 7.2. Methods of OOD Generalization

**Based on the invariance principle.** Inspired from the invariance principle [15,19], many methods are proposed by designing various loss to extract features to better satisfy the principle itself. IRMv1 [8] is the first objective to address this in an end-to-end way by adding a gradient penalty to the classifier. Following this work, Krueger et al. [9] suggest penalizing the variance of the risks, while Xie et al. [39] give the same objective but take the square root of the variance, and many other alternatives can also be found [40,41,42]. It is clear that all of these methods aim to find an invariant predictor. Recently, Ahuja et al. [12] found that for the classification problem, finding the invariant predictor is not enough to extract causal features since the features could include spurious information to make the predictor invariant across training environments, and they propose IB-IRM to address such a failure. Similar ideas to IB-IRM can also be found in the work [43,44], where different loss functions are proposed to achieve the same purpose. Specifically, Alesiani et al. [44] also use the information bottleneck (IB) for the help in dropping spurious correlations, but their analyses only focus on the scenario when spurious features are independent from the causal features, which could be considered as a special case of ours. More recently, Wang et al. [45] propose similar ideas to ours but only tackle the situation when the invariant features have the same distribution among all environments. In this work, we further show that IB-IRM could still fail in two cases due to the model only relying on spurious features to meet the task of interest. We then propose a counterfactual supervision-based information bottleneck (CSIB) method to address such failures and show improving results to prior works.

**Based on distribution matching.** It is worth noting that there are many works focused on learning domain invariant features representations [46,47,48]. Most of these works are inspired by the seminal theory of domain adaptation [49,50]. The goal of these methods is to learn a feature extractor Φ such that the marginal distribution of P(Φ(X)) or the conditional distribution of P(Φ(X)|Y) is invariant across different domains. This is different from the invariance principle, where the goal is to make P(Y|Φ(X)) (or E(Y|Φ(X))) invariant. We refer readers to the papers of [8,51] for better understanding the details of why these distribution-matching-based methods often fail to address OOD generalization.

**Others.** Other related methods are varied, including by using data augmentation in both image level [52] or feature level [53], by removing spurious correlations through stable learning [54], and by utilizing the inductive bias of neural networks [3,55], etc. Most of these methods are empirically inspired from experiments and are verified on some specific datasets. Recently, empirical studies in [56,57] notice that the real effects of many OOD generalization (domain generalization) methods are weak, which indicates that the benchmark-based evaluation criteria may be inadequate to validate the OOD generalization algorithms.

## 8. Conclusions, Limitations and Future Work

In this paper, we focus on the OOD generalization problem of linear classification. We first revisit the fundamental assumptions and results of prior works and show that the condition of invariant features supporting overlap is not necessarily needed for the success of OOD generalization and thus propose a weaker counterpart. Then, we show two failure cases of IB-IRM (as well as ERM, IB-ERM, and IRM) and illustrate its intrinsic causes by theoretical analysis. We further propose a new method—counterfactual supervision-based information bottleneck (CSIB)—and theoretically prove its effectiveness under some weaker assumptions. CSIB works even when accessing data from a single environment and can easily extend to the multi-class problems. Finally, we design several synthetic datasets with our examples for experimental verification. Empirical observations among all comparing methods illustrate the effectiveness of the CSIB.

Since we only take the linear problem into account, including linear representation and linear classifier, any nonlinear case would not be guaranteed by our theoretical results, and thus CSIB may fail. Therefore, the same as prior works (IRM [8] and IB-IRM [12]), the nonlinear challenge is still an unsolved problem [21,22]. We believe this is of great value for investigating in future work since widely used data in the wild are nonlinearly generated. Another fruitful direction is to design a powerful algorithm for entropy minimization during the learning process of CSIB. Currently, we use the variance of features to replace the entropy of the features during optimization. However, variance and entropy are essentially different. A truly effective entropy minimization is the key to the success of CSIB. Another limitation of our method is that we have to require further supervision to the counterfactual examples during the learning process, although it only takes one time for a single step. 

## Figures and Tables

**Figure 1 entropy-25-00193-f001:**
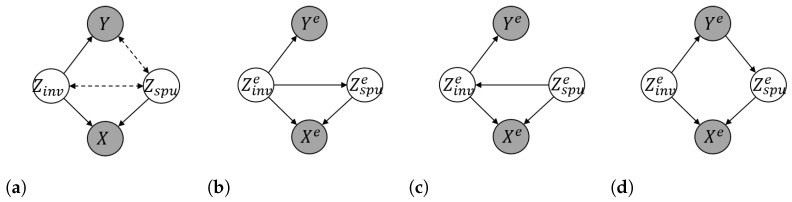
(**a**) DAG of the SEM $C_{ood}$ (Assumption 1); (**b**–**d**) DAGs of the interventional SEM $C_{ood}^{e}$ in the training environments $E_{tr}$ with respect to different correlations between $Z_{inv}$ and $Z_{spu}$. Grey nodes denote observed variables, and white nodes represent unobserved variables. Dashed lines denote the edges which might vary across the interventional environments and even be absent in some scenarios, whilst solid lines indicate that they are invariant across all the environments. All exogenous noise variables are omitted in the DAGs.

**Figure 2 entropy-25-00193-f002:**
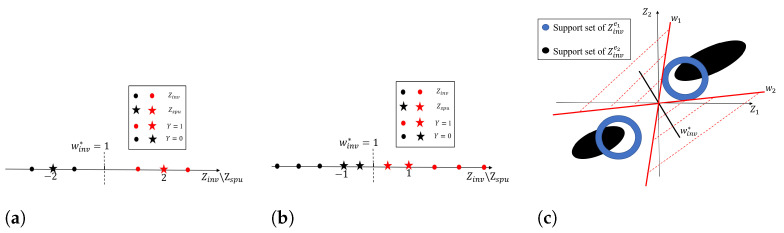
(**a**) Example 1; (**b**) example 2; (**c**) example illustration. Here, $dim(Z_{inv})=2$ and $Z_{inv}=\left(Z_{1},Z_{2}\right)$. The blue and black regions represent the support sets of $Z_{inv}^{e_{1}}$ and $Z_{inv}^{e_{2}}$, corresponding to the environments $e_{1}$ and $e_{2}$, respectively. $E_{tr}=\left\{e_{1}\right\}$ is the training environment and $E_{ood}=\left\{e_{2}\right\}$ is the OOD environment. Although Assumption 7 does not hold in this example, any zero-error classifier with $\Phi \left(X\right)=Z_{inv}$ on the $e_{1}$ environment data would clearly make the classification error zero in $e_{2}$, thus succeeding to solve OOD generalization.

**Figure 3 entropy-25-00193-f003:**
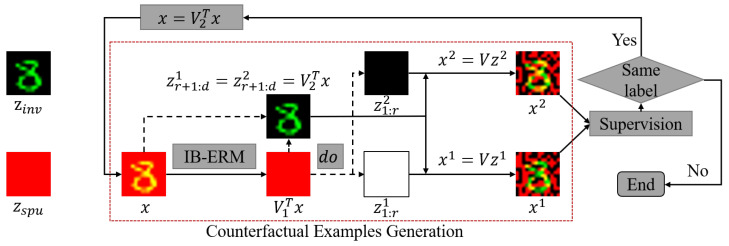
A simplified framework for the illustration of the proposed CSIB method.

**Figure 4 entropy-25-00193-f004:**
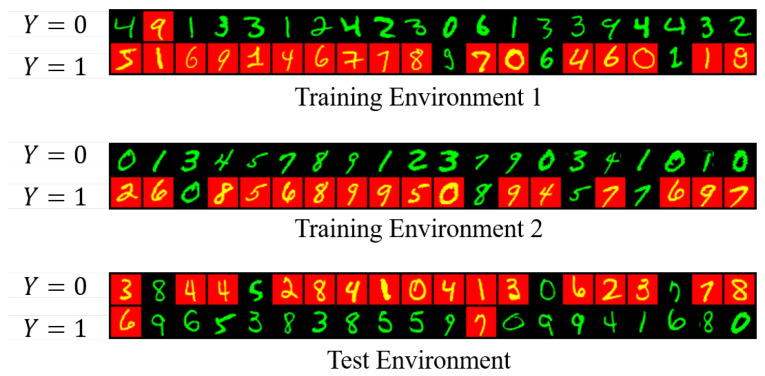
Visualization of the color mnist dataset.

**Table 1 entropy-25-00193-t001:** Summary of three synthetic datasets. Note that for linearly separable features, their margin levels significantly influence the final learning classifier due to the implicit bias of the gradient descent [18]. Such bias would push the standard learning (such as cross-entropy loss) to focus more on the large-margin features. The margin with respect to a dataset (or features) Z (each instance has a label 0 or 1) is the minimum distance between a point in Z and the max-margin hyperplane, which separates Z by its labels.

Datasets	Margin Relationship	Entropy Relationship	${\mathrm{Dim}}_{\mathrm{inv}}$	${\mathrm{Dim}}_{\mathrm{spu}}$
Example 1/1S	${\mathrm{Margin}}_{inv}\ll {\mathrm{Margin}}_{spu}$	${\mathrm{Entropy}}_{inv}<{\mathrm{Entropy}}_{spu}$	5	5
Example 2/2S	${\mathrm{Margin}}_{inv}\approx {\mathrm{Margin}}_{spu}$	${\mathrm{Entropy}}_{inv}>{\mathrm{Entropy}}_{spu}$	5	5
Example 3/3S	${\mathrm{Margin}}_{inv}\gg {\mathrm{Margin}}_{spu}$	${\mathrm{Entropy}}_{inv}>{\mathrm{Entropy}}_{spu}$	5	5

**Table 2 entropy-25-00193-t002:** Main results: #Envs means the number of training environments, and (min) reports the minimal test classification error across different running seeds.

	#Envs	ERM (min)	IRM (min)	IB-ERM (min)	IB-IRM (min)	CSIB (min)
Example 1	1	0.50 ± 0.01 (0.49)	0.50 ± 0.01 (0.49)	**0.23 ± 0.02 (0.22)**	0.31 ± 0.10 (0.25)	**0.23**± 0.02 (0.22)
Example 1S	1	0.50 ± 0.00 (0.49)	0.50 ± 0.00 (0.50)	0.46 ± 0.04 (0.39)	**0.30 ± 0.10 (0.25)**	0.46 ± 0.04 (0.39)
Example 2	1	0.40 ± 0.20 (0.00)	0.50 ± 0.00 (0.49)	0.50 ± 0.00 (0.49)	0.46 ± 0.02 (0.45)	**0.00 ± 0.00 (0.00)**
Example 2S	1	0.50 ± 0.00 (0.50)	0.31 ± 0.23 (0.00)	0.50 ± 0.00 (0.50)	0.45 ± 0.01 (0.43)	**0.10 ± 0.20 (0.00)**
Example 3	1	0.16 ± 0.06 (0.09)	0.18 ± 0.03 (0.14)	0.50 ± 0.01 (0.49)	0.40 ± 0.20 (0.01)	**0.11 ± 0.20 (0.00)**
Example 3S	1	0.17 ± 0.07 (0.10)	**0.09 ± 0.02 (0.07)**	0.50 ± 0.00 (0.50)	0.50 ± 0.00 (0.50)	0.21 ± 0.24 (0.00)
Example 1	3	0.45 ± 0.01 (0.45)	0.45 ± 0.01 (0.45)	**0.22 ± 0.01 (0.21)**	0.23 ± 0.13 (0.02)	**0.22 ± 0.01 (0.21)**
Example 1S	3	0.45 ± 0.00 (0.45)	0.45 ± 0.00 (0.45)	0.41 ± 0.04 (0.34)	**0.27 ± 0.11 (0.11)**	0.41 ± 0.04 (0.34)
Example 2	3	0.40 ± 0.20 (0.00)	0.50 ± 0.00 (0.50)	0.50 ± 0.00 (0.50)	0.33 ± 0.04 (0.25)	**0.00 ± 0.00 (0.00)**
Example 2S	3	0.50 ± 0.00 (0.50)	0.37 ± 0.15 (0.15)	0.50 ± 0.00 (0.50)	0.34 ± 0.01 (0.33)	**0.10 ± 0.20 (0.00)**
Example 3	3	0.18 ± 0.04 (0.15)	0.21 ± 0.02 (0.20)	0.50 ± 0.01 (0.49)	0.50 ± 0.01 (0.49)	**0.11 ± 0.20 (0.00)**
Example 3S	3	0.18 ± 0.04 (0.15)	0.08 ± 0.03 (0.03)	0.50 ± 0.00 (0.50)	0.43 ± 0.09 (0.31)	**0.01 ± 0.00 (0.00)**
Example 1	6	0.46 ± 0.01 (0.44)	0.46 ± 0.09 (0.41)	**0.22 ± 0.01 (0.20)**	0.37 ± 0.14 (0.17)	**0.22 ± 0.01 (0.20)**
Example 1S	6	0.46 ± 0.02 (0.44)	0.46 ± 0.02 (0.44)	**0.35 ± 0.10 (0.23)**	0.42 ± 0.12 (0.28)	**0.35 ± 0.10 (0.23)**
Example 2	6	0.49 ± 0.01 (0.48)	0.50 ± 0.01 (0.48)	0.50 ± 0.00 (0.50)	0.30 ± 0.01 (0.28)	**0.00 ± 0.00 (0.00)**
Example 2S	6	0.50 ± 0.00 (0.50)	0.35 ± 0.12 (0.25)	0.50 ± 0.00 (0.50)	0.30 ± 0.01 (0.29)	**0.20 ± 0.24 (0.00)**
Example 3	6	0.18 ± 0.04 (0.15)	0.20 ± 0.01 (0.19)	0.50 ± 0.00 (0.49)	0.37 ± 0.16 (0.16)	**0.01 ± 0.01 (0.00)**
Example 3S	6	0.18 ± 0.04 (0.14)	**0.05 ± 0.04 (0.01)**	0.50 ± 0.00 (0.50)	0.50 ± 0.00 (0.50)	0.11 ± 0.20 (0.00)

**Table 3 entropy-25-00193-t003:** Classification accuracy (%) on color MNIST dataset. “Oracle” in the table means that the training and test data are in the same environment.

Methods	ERM	IRM	IB-ERM	IB-IRM	CSIB	Oracle
Accuracy	9.94 ± 0.28	20.39 ± 2.76	9.94 ± 0.28	43.84 ± 12.48	**60.03 ± 1.28**	84.72 ± 0.65

## Data Availability

Data is contained within the article or supplementary material.

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
