# Peer review of "Counterfactual Supervision-Based Information Bottleneck for Out-of-Distribution Generalization"

_entropy, 2023, doi:10.3390/e25020193_

Round 1

Reviewer 2 Report

The paper needs to be improved before publication

1. the failure of the method is lacking clarity

2. the method proposed is not well presented

3. experiments only include synthetic data

Details comments:

- "yet remain" in the abstract is vague, in general is better not to put reference in the abstract. Please be specific

- Introduction: "when domain distributions of testing ...": is the input domain? it is not clear which conditions machine learning fails

- line 58: 

            - calligraphic F is not defined 

            - "but related" is not clearly defined what it means

- Assumption 1 (line 62-64)

            - Zinv, Zspu orthogonality to N is not well defined (between random variable and a pair of vectors)

            - S is not defined

- line 67: not clear what orthogonality again means, please also provide explanation.

- line 71: "Zinv, Zspu depend on environment e" unclear defined, vague

- line 72: : "therefore" why? is not explained (and not true in general)

- Figure.1: X^e not defined in the caption description

- Figure 2(c): not clear what is visualized, also in the text after.

- line 78-80: give reference to support the claim

- Assumption 5: definition is not clear and its meaning too

- In general line 85-97, the theorem is not well presented.

- lines 101-104: not clear, please give a concrete example.

- lines 105-106: vague statement "traditional generalization theory".

- line 112-115: clarify the various elements used

- line 135: sentence is not correct. "only" is not referred to IB-IRM, but to Th.2

- Example.1 : the relationship is inverted between Z_inv and Z_spu, this is a different causal model

            - please specify o,m are the size of the variables

- equation (5): derivation not clear, at least add to Annex

- Line 141: "therefore": is not clear, why?

- Failure modes, which is a key contribution of the paper need to be clarified, not possible to follow otherwise the rest of the paper

- Additional comments:

- Section 4: the method is not well introduced, the steps are clear, but the description of what they intend to do it is not clear

- Eq.8: will not this fail as described before?

- Please explain the aim of step 2

- Step 3: why setting part to M, why is important to have M? how M is computed?

- Table 2: the values presented are not well defined

            - please highlight the best method in bold

- General comment: the work present experiments on generated data, please provide at least one experiment with some realistic data

-  In addition to [4] that combines IB and IRM, the work [1] below also consider the relationship between IRM and IB in sequential environments

[1] Alesiani, Francesco, Shujian Yu, and Xi Yu. "Gated Information Bottleneck for Generalization in Sequential Environments." In 2021 IEEE International Conference on Data Mining (ICDM), pp. 1-10. IEEE, 2021.

Round 2

Reviewer 1 Report

Thanks for addressing my concerns.

Author Response

We would like to thank the reviewer agian for providing helpful comments to improve our paper.

Reviewer 2 Report

I checked the changes and the author reply. Unfortunately I am not able to read for the first time to understand if the content is presented appropriately, but the reply given are fine with me, please make sure that the explanation are somehow considered in the main text. The replies you gave to me are for any reader. 

Author Response

Thanks for the comments again. All explanations about the concerns are included in the main text of the manuscript. 

Reviewer 3 Report

In this revision, the authors have embedded their approach into the structural causal model framework of Pearl and have formalized their definition of "success" in OOD generalization.  Having said that, I actually now believe the paper is based on a faulty premise.   The paper attempts to use invariant causal predictors and talks about failures of OOD when there is lack of overlap in the support of the invariant causal predictors.   My question to the authors is how likely is this to happen?  Specifically, this would be there would no overlap of the invariant causal predictors in the training and test datasets.  This doesn't make sense to me.  

I also think that the results of Barenboim and Pearl regarding data fusion would apply here and that in fact, the problem the authors treat and the results they find run in contrast to Barenboim and Pearl. 
